# Variation of groundwater and mineral composition of in situ leaching uranium in Bayanwula mining area, China

Haibo Li[1], Akhtar Malik Muhammad[2]*, Zhonghua Tang[1]

**1** School of Environmental Studies, China University of Geosciences, Wuhan, Hubei Province, China PRC, **2** Department of Environmental Science, Faculty of Life Sciences and Informatics, Balochistan University of Information Technology, Engineering and Management Sciences, Quetta, Pakistan

* drmalikma21@gmail.com

## Abstract

The reaction between the lixiviant and the minerals in the aquifer of In-situ uranium leaching (ISL) will result mineral dissolution and precipitation. ISL will cause changes in the chemical composition of groundwater and the porosity and permeability of aquifer, as well as groundwater pollution. Previous studies lack three-dimension numerical simulation that includes a variety of minerals and considers changes in porosity and permeability properties simultaneously. To solve these problems, a three-dimensional reactive transport model (RTM) which considered minerals, main water components and changes in porosity and permeability properties in Bayanwula mine has been established. The results revealed that: (1) Uranium elements were mainly distributed inside the mining area and had a weak trend of migration to the outside. The strong acidity liquid is mainly in the mining area, and the acidity liquid dissolved the minerals during migrating to the outside of the mining area. The concentration front of major metal cations such as $K^+$, $Na^+$, $Ca^{2+}$ and $Mg^{2+}$ is about 150m away from the boundary. (2) The main dissolved minerals include feldspar, pyrite, calcite, sodium montmorillonite and calcium montmorillonite. Calcite is the most soluble mineral and one of the sources of gypsum precipitation. Other minerals will dissolve significantly after calcite is dissolved. (3) ISL will cause changes in porosity and permeability of the mining area. Mineral dissolution raises porosity and permeability near the injection well. Mineral precipitation reduced porosity and permeability near the pumping well, which can plugging the pore throat and affect recovery efficiency negatively.

## 1. Introduction

Over half of the world population depends on groundwater for drinking and other domestic uses [1] and important for economic development [2]. Due to natural protection system, groundwater is considered contamination free water source and under-developing/developing nations must depend for domestic water supply with quality monitoring system [3]. Uranium is a radioactive element and is considered toxic for human body even in small quantity [4].

**Data Availability Statement:** This research paper is part of research project (support project NSFC (u1911205) and the Beijing Research Institute of Chemical Engineering and Metallurgy through the project A60-3). Interested researchers can contact

the Beijing Research Institute of Chemical
Engineering and Metallurgy to apply for data
access: 13932100832@163.com.

**Funding:** This study is supported partially by Joint
fund key support project NSFC (u1911205) and the
Beijing Research Institute of Chemical Engineering
and Metallurgy through the project A60-3. The
authors would like to thank Mr. Yiwei Li and Prof.
Jili Wen from Beijing Research Institute of
Chemical Engineering and Metallurgy for advising
the work. Editors and two anonymous reviewers
are also thanked for the valuable comments in
improving the quality of the manuscript.

**Competing interests:** The authors have declared
that no competing interests exist.

Various factors could be involved to elevate the uranium concentration in aquifers such as uranium mining, groundwater depletion, excessive use of phosphate fertilizers, etc [5].

ISL is an important method for uranium mining. In-situ leaching of uranium is a typical solute transport problem [6]. The whole process is under natural burial conditions, the lixiviant is injected into the ore bed through a liquid injection well, and the leaching solution contacts with the useful components in the ore to produce a chemical reaction. The resulting soluble compounds leave the chemical reaction zone under the action of diffusion and convection, migrate to the extraction pipe through infiltration, lift to the surface, process, and extract again, and finally obtain qualified products. Compared with open-pit mining and underground mining, this method has the advantages of low cost and less impact on the environment.

However, since in ISL, reactions between lixiviant and minerals not only release uranium but also dissolve many minerals and cause other minerals precipitation [7–10]. During acid ISL production, due to the precipitation of insoluble minerals, the pore-plugging of sandstones is commonly encountered. The decreasing of permeability constrains the lixiviant flow and lowers the uranium recovery rate. Alteration of minerals like feldspar and clay minerals is the key factor controlling pore-plugging of sandstone in ISL. Currently, gypsum precipitated in ISL has attracted much attention [11], whereas other pore plugging types induced by the alteration or precipitation of minerals like feldspar and clay minerals are still less concerned. The dissolution and precipitation of minerals play an important role in ISL.

To the complex chemical composition in the leaching solution of uranium in situ leaching project, the groundwater will be polluted if it is not properly controlled. The process of uranium migration includes complex hydrogeochemical processes such as oxidation-reduction, solution-precipitation, complex-dissociation, adsorption-desorption, etc [12]. Therefore, it presents a "rolling" migration of solution-migration-sedimentation-resolution. The complex groundwater dynamic field and hydrogeochemical process control the speed, intensity, and state of solute transport, while the solute transport and its concentration change reveal the hydrodynamic characteristics and hydrogeochemical process of the in-situ leaching system [13, 14]. Many researchers have studied the transport characteristics and composition changes of solute during in situ leaching. These studies are of great significance for understanding the solute migration, acidification and oxidation of ore beds and the interaction between water and rock during in situ leaching. At the same time, it can guide the optimization of process parameters and improve production efficiency [15, 16]. Many researchers have studied the adsorption behavior of hexavalent uranium. Their research shows that hexavalent uranium is usually not easy to adsorb in a lower pH environment, while in the range of 4 to 6, the adsorption behavior of hexavalent uranium will be strengthened with the increase of pH [17–20]. Steven B. Yabusaki focused on the adsorption behavior of uranium and believed that the adsorption constant was related to the concentration of uranium and the alkalinity of groundwater [21]. With the increase of uranium concentration and alkalinity, the adsorption constantly decreased. Domestic research in this area has also carried out a lot of work. Sun and other researchers conducted a precipitation experiment based on the composition of the acid in-situ leaching solution of a uranium mine in Inner Mongolia, configured a uranium-containing solution with a single impurity metal ion, and studied the precipitation behavior of uranium and impurity metal ions [22–25]. This study concluded the precipitates of $Al(OH)_3$ and $Fe(OH)_3$ accelerating the precipitation of uranium, and the precipitation of uranium in the leaching liquid was greatly affected by Al. Mikhail Panfilov developed an asymptotic analytical solution of one dimensional flow problem with several chemical reactions to analysis the efficiency of ISL. In this research, the liquid components contain sulphuric acid, ferric sulphate, uranyl sulphate, ferrous sulphate, carbonic acid and water. The solid components

contain uraninite, uranium dioxide, ferric hydroxide calcium carbonate and gypsum (anhydrite) [26]. The method can obtain the analytical solutions of one-dimensional case. The result can be used to test the validity of numerical solvers and solve the inverse problem. In this research, the analytical solutions is only for one dimension case. The feldspar and clay minerals is not considered sufficient. Rose Ben Simon simulated the batch experiments with geochemical code CHESS and calibrate the geochemical reaction and their kinetic laws [27]. And then simulated the column experiments with the coupled hydrodynamic and geochemical code HYTRC by using the react laws calibrated on batch experiment. The geochemical models described the chemistry of ISL in the batch experiment and column experiment. The sensitivity analysis of the one-dimensional column model revealed that the determine factors controlling the ISL reactions and quantify their respective influence on the ISL in terms of acid consumption and leachate volume. In this research, the geochemical model contained uraninite, schoepite, feldspar, calcite, dolomite, pyrite, hematite and several other minerals, but not consider the clay minerals. And the model is applicable to batch experiment case or column case. Antoine Collet adopt three-dimensional reactive transport model simulated the ISL and optimized using a geometallurgical approach [28]. The geochemical model relies on six minerals: cristobalite alpha, kaolinite, smectites (beidellite), calcite, iron hydroxides (goethite), and uraninite. The minerals need to be considered more fully.

Bayanwula mining area has a relatively complete mining history. There have been some studies carried out in this area. Zheng and Chen established a hydrodynamic model for ISL in the study area with VisualModflow, and studied the effects of different pumping and injection fluid flow rates and pumping hole distances on the percolation characteristics of the solution during in-situ leaching [29, 30]. In these studies, the geochemical reaction is not considered. Jiao Youjun used the PHT3D software to analyze the effect of different adsorption models on the behavior of hexavalent uranium in groundwater by numerical simulation [31, 32]. Chen used PHT3D to carry out numerical simulation research on the influence of calcite and pyrite minerals in pitch-bearing uranium mining areas [33]. In this research, the simulation used two dimension model, and the minerals contains uranium ore, calcite, pyrite and hematite.

In Bayanwula mining area, the ore-bearing aquifer is tens of meters thick, while the thickness of the ore body is generally within a few meters. The wells are partially penetrating well. The ore-bearing aquifer contains uraninite, feldspar, calcite, hematite and clay minerals. The dissolution and precipitation of minerals are significant, and the variation in the porosity and permeability of the aquifer is obvious. One-dimensional model and two-dimensional plane model were used in the ISL simulation of Bayanwula mining area. These models treat ore-bearing aquifers as one layer and wells as fully penetrating well, therefore they cannot characterize the vertical flow of groundwater in ISL. The numerical simulation work containing hydrogeochemistry only takes uranium ore, calcite, pyrite and hematite as the mineral set for simulation study, and the influence of fully containing minerals in the ore layer on in-situ uranium leaching is still lacking. Due to the limitation of the software function, the previous numerical simulation of in-situ uranium leaching in mining area did not calculate the change of porosity and permeability of ore-bearing aquifer.

In this study, a three-dimensional reactive solute transport model was used for simulation. In the model, the wells are treated as partially penetrating well, and the mineral components and hydrogeological parameters are zoned horizontally and vertically. These methods can more accurately depict the effects of vertical groundwater flow. For mineral components, all the above minerals in the mining area are considered in this study, which makes the mineral transformation relationship and hydrogeochemical characteristics in the mining more accurate. At the same time, the change of porosity and permeability of ore-bearing aquifer caused by mineral dissolution or precipitation is calculated.

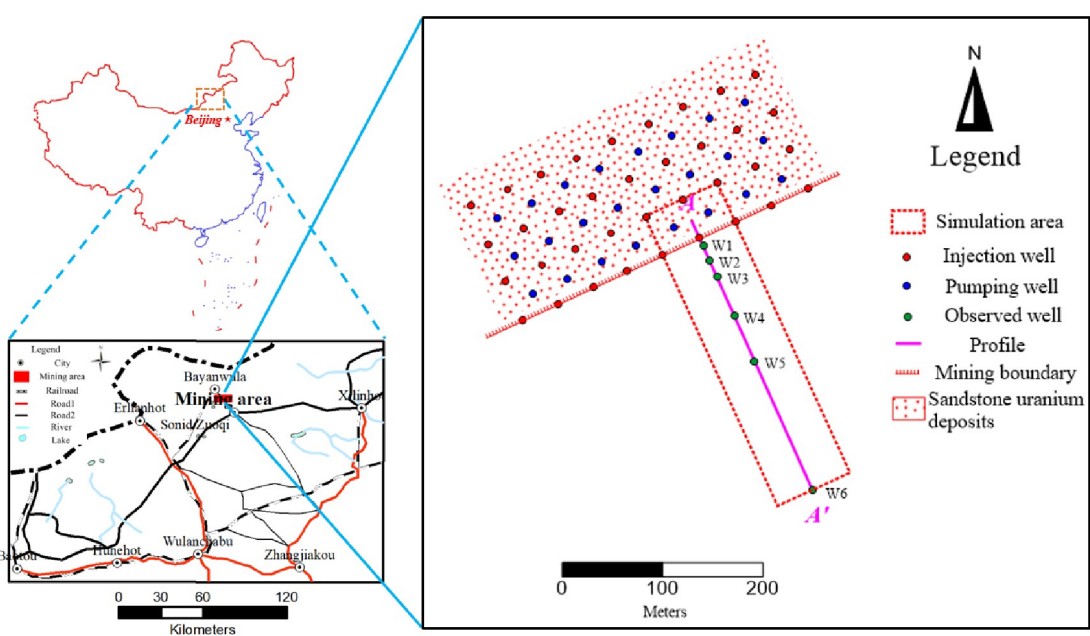

**Fig 1. Traffic location map of Bayanwula mining area.**

## 2. Study area

The Bayanwula mining area is in the northeast of Inner Mongolia Autonomous Region and the northwest of Sonid Zuoqi, with a small overall area of about 90km$^2$. Fig 1 shows the specific location of the proposed site with other informations.

### 2.1 Hydrogeological conditions of the study area

The exposed strata of the mine and nearby surface are the Paleogene Irdimanha group, which is largely covered by the Quaternary ($Q_4$) system, and the Saihan group has no outcrop. According to the borehole lithology data, the strata exposed in the mine from bottom to top include the lower section of the Saihan group ($K_1s^1$), the upper section of the Saihan group ($K_1s^2$), and the Paleogene Irdimanha group ($E_2y$) (Fig 2).

The clastic rock aquifer subgroup ($E_2y$) of the Irdimanha Formation was distributed throughout the test area and covered by the Quaternary system gradually thickness in the south and west with value 40 meters to 60 meters while thickness in the northeast is 60 meters to 99 meters. The study area contained a variety of sedimentary sandstone, sandy conglomerate, argillaceous sandy conglomerate, sandy mudstone, mudstone, etc.

The aquifer subgroup ($K_1s$) of the clastic rock of the Saihan Formation distributed in the area and buried under the Irdimanha Formation while consists of the upper section of the Saihan Formation ($K_1s^2$) and the lower section of the Saihan Formation ($K_1s^1$) contained pore confined water. The upper section of Saihan Formation was mainly composed of sandstone, sandy conglomerate mixed with mudstone and silty mudstone mined by braided river with various thickness of 40m to 100m, 33.7m and 107.0m. Among the sandstone and sandy conglomerate in the middle and lower parts were loose in structure, with good connectivity, gentle occurrence, stable and continuous distribution, good water yield and permeability with a total thickness of 30m to 90m forming an ore aquifer. However, the water level buried depth of 13m

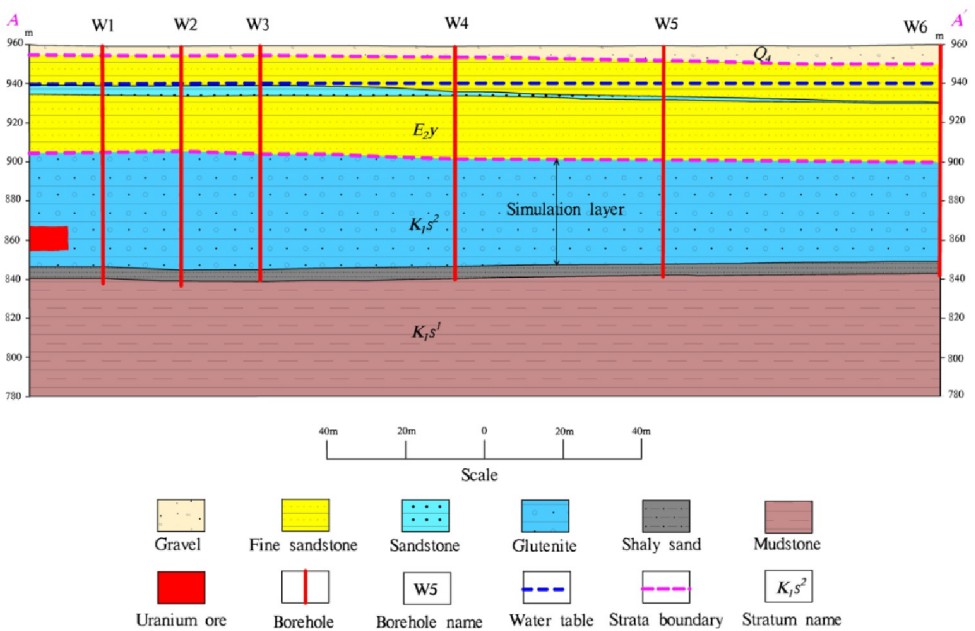

**Fig 2. Typical section of the study area.**

to 40m, and water inflow into the well greater than 100m³/d. The mudstone thick at top was observed 5m to 20m and distributed forming a stable water-resisting roof of the ore aquifer. The lower section of Saihan Formation is composed of mudstone and silty mudstone mixed with lignite mined in the lake and marsh. The thickness of borehole exposure was generally 5 meters to 60m and stable water-resisting floor of the aquifer. Fig 2 shows typical hydrogeological section of the study area. The main aquifer is located at K1 stratum, and the aquifer is layered. The stratigraphic structure was relatively accurate, and the water head was very gentle, which was taken as 941m through actual measurement [29, 30].

## 3. Model establishment

The initial conditions and boundary conditions of the model apply the initial pressure setting of the water pressure was combined with the hydrogeological conditions of the mining area. The values were computed according to the water head. Due to the low hydraulic gradient of the mining area, groundwater movement was slow. The water head distribution was relatively flat. The initial water head at the top of the aquifer was almost 941m. The boundary conditions were adjusted based on the field visit. It was observed that the water head 300m away from the mining area was not affected by mining. Therefore, the observation points 300m away from the mining site were selected as the boundary of the model. Polygonal grids were used in the plane while rectangular grids are selected in the vertical for grid division, which were more conducive to describing the transport of reactive solute near the wellbore. The model's detailed description is given below.

### 3.1 Conceptual model

**3.1.1 Simulation area.** The simulation area was done at the end point of the mining area, that is, from the two parallel pumping and injection units at the most end point of the selected area. The pumping and injection well have distance of almost 300m and installed five

observation wells in the simulation area. The division and selection of the simulation area were mainly based on the following:

(1) Based on the distribution characteristics of the flow field around the mining area. The distribution of pumping and injection wells was very tidy. So, the inflow and outflow must be same from the injection and pumping well. Moreover, the actual investigation and numerical simulation study of similar mining models showed that the flow field formed by such mining areas is characterized by parallel flow gradually outward from the mining area boundary, and the water level lines are approximately parallel to each other and gradually increasing. Therefore, the two long lateral boundaries on both sides of the simulation area were treated as impermeable boundaries. (2) The groundwater flow in the study area was very small, and the change of local flow field is mainly controlled by the mining wells.

(3) Each production area was pumped in the mode of four injections wells and one pumping well. The flow field developed by the pumping well was the convergence of four injection wells to one pumping well, which was like the four-leaf petal-shaped distribution. In the direction of the connecting line of the injection wells, it could be approximated as the water barrier boundary.

(4) There was an observation well located 300m away from the mining area. According to the value of the water table at the observation well, the fluctuation in water table was slightly affected due to activities at the mining area. Therefore, at the distance of 300m, it could be regarded as a fixed water head boundary to control the pressure boundary.

Based on the above analysis, the boundary of the simulation area was categorized into three impermeable boundaries and one fixed head boundary shown in Fig 3. The orange area is a mining area and has two pumping and one injection wells while the white area is the periphery of the mining area, without ore. The red point in the figure represents the injection well, and the blue dot represents the pumping well.

**3.1.2 Parameter zoning.** Parameter zoning mainly included permeability zoning and mineral composition zoning. Due to the little change of initial permeability within the region in this simulation, the initial permeability in the simulation zone remains the same, while mineral composition was divided into uranium and non-uranium zones. According to the mining area scope in the model generalization (Fig 4), the ore and ore-free zones were represented with different colors. With reference to the regional geological report, the spatial location

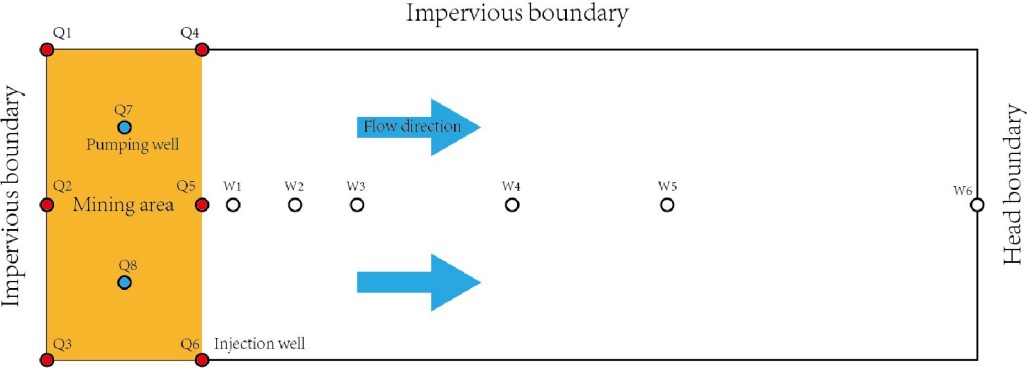

**Fig 3. Conceptual model of reactive transport.**

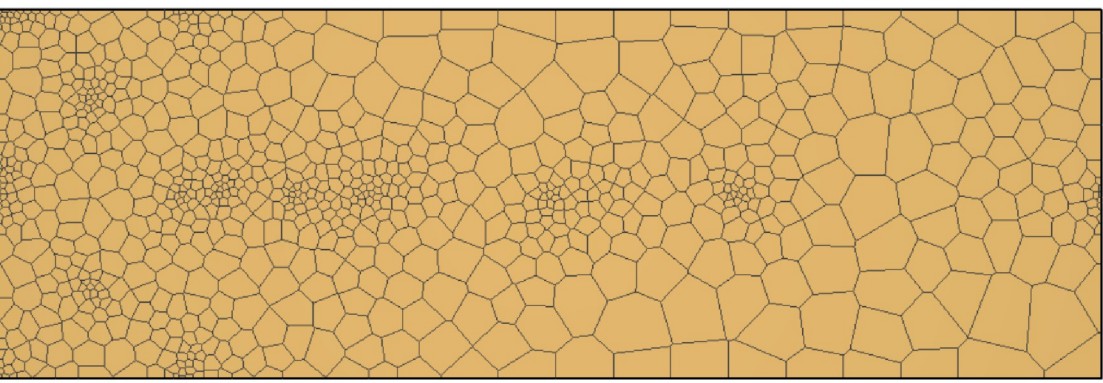

**Fig 4. Plane diagram of mesh.**

partition and the initial parameter setting of the model was verified to the mineral component information.

## 3.2 Mathematical model

**3.2.1 Equations.** From the perspective of subsurface flow system process, the migration of reactive solute is actually the coupling of convective dispersion and hydrogeochemical reaction. Based on the conservation of mass and energy and Darcy's law, a mathematical model of groundwater flow can be established. Mass conservation equations of material components (groundwater, salt and gas components) are adopted. Through this method, a multiphase system can be well described, and the transformation of the same substance between different phase states can also be described. In this process, it is necessary to consider the thermodynamic properties of the phase states and components. The governing equation involved in the model is as follows [34]:

(1) The equation can be established from the conservation of mass:

$$\frac{\partial M_\kappa}{\partial t} = -\nabla F_\kappa + q_\kappa \tag{1}$$

$$F_h = \sum_{\beta=l,g} h_\beta \rho_\beta \overrightarrow{u_\beta} - \lambda \nabla T, q_h \tag{2}$$

The mass flux of the multi-phase is according to Darcy's law:

$$\overrightarrow{u_\beta} = -k \frac{k_{r\beta}}{\mu_\beta} \left( \nabla P_\beta - \rho_\beta \vec{g} \right) \tag{3}$$

where, $\overrightarrow{u_\beta}$ is the mass flux of phase $\beta$, $\rho_\beta$ is the density of phase $\beta$, $k$ is the absolute permeability of the rock, $k_{r\beta}$ is the relative permeability of phase $\beta$, $\mu_\beta$ is viscosity, and $P_\beta$ is the fluid pressure of phase $\beta$. $\vec{g}$ is the vector of gravitational acceleration.

(2) Reactive transport model

In the mathematical model, chemical reactions mainly occur in the liquid phase. For chemical components in the liquid phase:

$$F_j = c_{jl}\overrightarrow{u_l} - \tau\varphi S_l D_l \nabla c_{jl}, q_j = q_{jl} + q_{jg} + q_{jr} \tag{4}$$

$$M_j = \varphi S_l c_{jl}, \tau = \varphi^{1/3} S_l^{7/3} \tag{5}$$

where, $F_j$ is the mass flux of species $j$, $\overrightarrow{u_l}$ is the mass flux of liquid phase, $M_j$ is the mass of the species $j$ in the liquid phase, $c_{jl}$ is the concentration of the species $j$ in the liquid phase, $\tau$ is the tortuosity, $\varphi$ is the porosity, $S_l$ is the liquid phase saturation, $D_l$ is the hydrodynamic dispersion coefficient, $q_j$ is the source and sink.

Combine with the mass conservation equation, which can be expressed as:

$$\frac{d\left(\varphi S_l c_{jl}\right)}{dt} = -\nabla\left(c_{jl}\overrightarrow{u_l} - \tau\varphi S_l D_l \nabla c_{jl}\right) + \sum_{i=1}^{N} v_{jr} r_n \tag{6}$$

where $v_{jr}$ is the stoichiometric number of species $j$ in the $r$th reaction. This parameter was form the mineral reaction equations. $r_n$ is the reaction rate.

The main chemical reaction mathematical models in the simulation include equilibrium mineral and kinetic mineral. Equilibrium mineral model is:

$$\Omega_m = K_m^{-1} \prod_{j=1}^{N_c} c_j^{v_{mj}} \gamma_j^{v_{mj}} \tag{7}$$

where $K_m$ is the equilibrium constant.

Kinetic mineral model is:

$$r_n = \pm k_n A_n |1 - \Omega_n^{\theta}|^{\eta} \tag{8}$$

where $A_n$ is the specific surface area of the mineral; $k_n$ is the nth parallel mineral precipitation or dissolution reaction rate constant that depends on temperature;

Based on the Arrhenius equation, the correlation between $k_n$ and temperature can be written as:

$$k_n = k_{25}^{nu} exp\left[-\frac{E_a^{nu}}{R}\left(\frac{1}{T} - \frac{1}{298.15}\right)\right] + \sum_i\left\{k_{25}^i exp\left[-\frac{E_a^i}{R}\left(\frac{1}{T} - \frac{1}{298.15}\right)\right]\right\}\prod_j a_{ij}^{n_{ij}} \tag{9}$$

where, $\prod_j a_{ij}^{n_{ij}}$ describes the effect of specific ion activity on the i th parallel mineral precipitation and dissolution reaction;

(3) Mineral reaction equations

(a) Uranium minerals

The uranium mineral is $UO_2$, and the reaction equation is as follows:

$$2UO_2 + 4H^+ + O_2 = 2UO_2^{2+} + 2H_2O \tag{10}$$

(b) Secondary minerals

Based on the geological condition, the secondary minerals include gypsum and anhydrite, hematite, muscovite, dolomite, siderite, and ankerite. The reaction of these minerals are discussed in followings:

1) Gypsum and anhydrite

In acid ISL production, when high-acidity leaching solution is injected into the ore layer through the injection hole, sulfuric acid will react with limestone in the ore layer, produce anhydrite ($CaSO_4$) or gypsum ($CaSO_4 \cdot 2H_2O$) precipitation. The reaction equation is as follows:

$$CaCO_3 + 2H^+ = Ca^{2+} + H_2O + CO_2(g) \tag{11}$$

$$Ca^{2+} + SO_4^{2-} + 2H_2O = CaSO_4 \cdot 2H_2O(s) \tag{12}$$

2) Hematite

Hematite is also the main mineral that produces chemical precipitation in the ISL area of Bayanwula uranium mine. When there is a certain amount of $Fe^{3+}$ in the water, it is also easy to form hematite precipitation. The reaction formula is as follows:

$$\mathbf{2Fe^{3+} + 3H_2O = Fe_2O_3 + 6H^+} \tag{13}$$

In addition, some other minerals such as muscovite, dolomite, siderite, and ankerite, are also found in the study area. These minerals are also considered in the model. Their chemical reaction formulas are displayed in Table 1.

Relative permeability and capillary pressure calculation model

The calculation model of relative permeability and capillary pressure adopts Van Genuchten-Mualem (Van Genuchten, 1980) model.

The calculation model of relative permeability and capillary pressure in the model adopted by Van Genuchten-Mualem model [35]. The calculation formula was used shown in the Table 2.

**3.2.2 Initial condition.** *1. Initial temperature and initial head.* According to the results of the water quality analysis of the mining area, the initial temperature of the aquifer is 9℃. The aquifer of the simulation is in the shallow layer. The regional geological survey and the research area have no strong geothermal gradient. Therefore, the effect of temperature variations on the system is not considered in this study. The whole process is an isothermal simulation. According to field research, the initial head has been set as 941m.

*2. Initial concentration of main ions in groundwater.* To determine the initial concentration of main ions in groundwater, a water sample was collected in the mining area, and the sample point was located in the upstream of the mining area and far from the mining area. According to the analysis result of the single sample, the initial concentration of ions is listed in Table 3:

*3. Initial minerals.* In Table 4, the initial mineral composition is divided into feldspar minerals, clay minerals, uranium minerals, iron minerals and carbonate minerals. The initial mineral composition of the ore layer can be generalized as follows:

**Table 1. Secondary mineral in simulation.**

| Mineral | Chemical formula |
|---------|------------------|
| Muscovite | $KAl_2(AlSi_3O_{10})(OH)_2$ |
| Dolomite | $CaMg(CO_3)_2$ |
| Siderite | $FeCO_3$ |
| Ankerite | $CaMg_{0.3}Fe_{0.7}(CO_3)_2$ |

**Table 2. Relative permeability and capillary pressure of the model.**

| Liquid relative permeability, use the Van Genuchten-Mualem model [35, 36]: | |
|---|---|
| $k_{rl} = \sqrt{S^*}\left\{1 - \left[1 - [S^*]^{1/\lambda}\right]^{\lambda}\right\}^2, S^* = (S_l - S_{lr})/(S_{ls} - S_{lr})$ | |
| $\lambda$:exponent | 0.457 |
| $S_{lr}$:residual gas saturation | 0.05 |
| $S_{ls}$ | 1 |
| Gas relative permeability, Corey [37]: | |
| $k_{rg} = \left(1 - \hat{S}\right)^2\left(1 - \hat{S}^2\right), \hat{S} = (S_l - S_{lr})/\left(1 - S_{lr} - S_{gr}\right)$ | |
| $S_{gr}$:residual gas saturation | 0.05 |
| Capillary pressure, use the Van Genuchten-Mualem model [35]: | |
| $P_{cap} = -P_0([S^*]^{-1/\lambda} - 1)^{1-\lambda}$ | |
| $S^* = (S_l - S_{lr})/(S_{ls} - S_{lr})$ | |
| $\lambda$ | 0.457 |
| $S_{lr}$, residual water saturation | 0.3 |
| $S_{ls}$ | 0.999 |
| $P_{max}$ | $1 \times 10^7$ |
| $P_0$ | $1.96 \times 10^4$ |

**Table 3. Initial concentration.**

| Index | Concentration |
|---|---|
| Na(mg/L) | 495 |
| K(mg/L) | 8.46 |
| Ca(mg/L) | 76.6 |
| Mg(mg/L) | 55.2 |
| $SO_4^{2-}$(mg/L) | 407 |
| $Cl^-$(mg/L) | 410 |
| $HCO_3^-$(mg/L) | 1110 |
| Fe(mg/L) | 0.747 |
| U(μg/L) | 30.1 |

**Table 4. Initial mineral composition.**

| name | Chemical formula | Initial mineral volume fraction of ore body | Initial mineral volume fraction of surrounding rock |
|---|---|---|---|
| Quartz | $SiO_2$ | 0.48686 | 0.48686 |
| K-feldspar | $KAlSi_3O_8$ | 0.21550 | 0.21550 |
| Oligoclase | $CaNa_4Al_6Si_{14}O_{40}$ | 0.01134 | 0.01134 |
| Na-smectite | $Ca_{0.145}Mg_{0.26}Al_{1.77}Si_{3.97}O_{10}(OH)_2$ | 0.08391 | 0.08391 |
| Ca-smectite | $Na_{0.29}Mg_{0.26}Al_{1.77}Si_{3.97}O_{10}(OH)_2$ | 0.02597 | 0.02597 |
| Illite | $K_{0.6}Mg_{0.25}Al_{1.8}(Al_{0.5}Si_{3.5}O_{10})(OH)_2$ | 0.02137 | 0.02137 |
| Kaolinite | $Al_2Si_2O_5(OH)_2$ | 0.01520 | 0.01520 |
| Uraninite | $UO_2$ | 0.01143 | 0.00000 |
| Hematite | $Fe_2O_3$ | 0.01880 | 0.01880 |
| Pyrite | $FeS_2$ | 0.00245 | 0.00028 |
| Calcite | $CaCO_3$ | 0.03320 | 0.01140 |

**Table 5. The concentration of injection well.**

| Index | Concentration |
|---|---|
| Na(mg/L) | 1950 |
| K(mg/L) | 533 |
| Ca(mg/L) | 441 |
| Mg(mg/L) | 916 |
| $SO_4^{2-}$(mg/L) | 27300 |
| $Cl^-$(mg/L) | 464 |
| $HCO_3^-$(mg/L) | 35 |
| Fe(mg/L) | 1580 |
| U(μg/L) | 0.79 |

**3.2.3 Boundary condition.** According to the conceptual model, there are three impermeable boundaries and one fixed head boundary. Based on the groundwater level at about 300m outside the mining area, the head of fixed head boundary is 941m.

**3.2.4 Source and sink.** *(1) Well flow rate.* The flow rate of injection wells is allocated according to the boundary of the model area and the distribution characteristics of the wells, if the injection well is at the edge of the boundary, half of the flow rate of a single well is taken; if the injection well is at the corner of the boundary, one quarter of the flow rate is taken. Since the pumping well is in the center of the pumping unit, the flow rate of a single well can be taken. According to the data of injection wells and production wells in the mining area, the flow rate of the injection well is 176m³/d, pumping well flow rate is 264m³/d.

*(2) The concentration of injection well.* To determine the chemical composition of liquid of injection wells in the simulation process, the injection well sample were taken from the injection wells. According to the analysis results of the single sample, the concentration of chemical components of injection well is as shown in Table 5.

## 3.3 Numerical model

**3.3.1 Mesh generation.** The distance between the injection wells was maintained almost 50m, and the pumping wells were in the middle of the study area and four injection wells were installed in its surrounding. There were six injection wells and two pumping wells installed to conduct this study. To better describe the underground water flow field and chemical field near the well, polygon grid was used for plane section and rectangular grid was used for vertical section which is divided into six layers. The fifth layer covered the mining area which represents the ore layer. The division is shown in Figs 4 and 5. The actual size of the grids in the east-west, north-south and vertical length is 300m, 100m and 60m, respectively. The vertical grid division information is shown in Table 6. The red part of the 3D grid is the ore bed (Fig 6).

**3.3.2 Aquifer parameters.** In this simulation, the initial permeability within the study area remains consistent. Based on the regional geological report, the spatial location information of the parameters, and the initial parameter setting of the model for the mineral component's information has been confirmed (Table 7).

**3.3.3 Thermodynamic and kinetic parameters of mineral reaction.** The mineral reaction mainly selected the dissolution and precipitation of relevant minerals, and its chemical reaction parameters are as follows the Table 8.

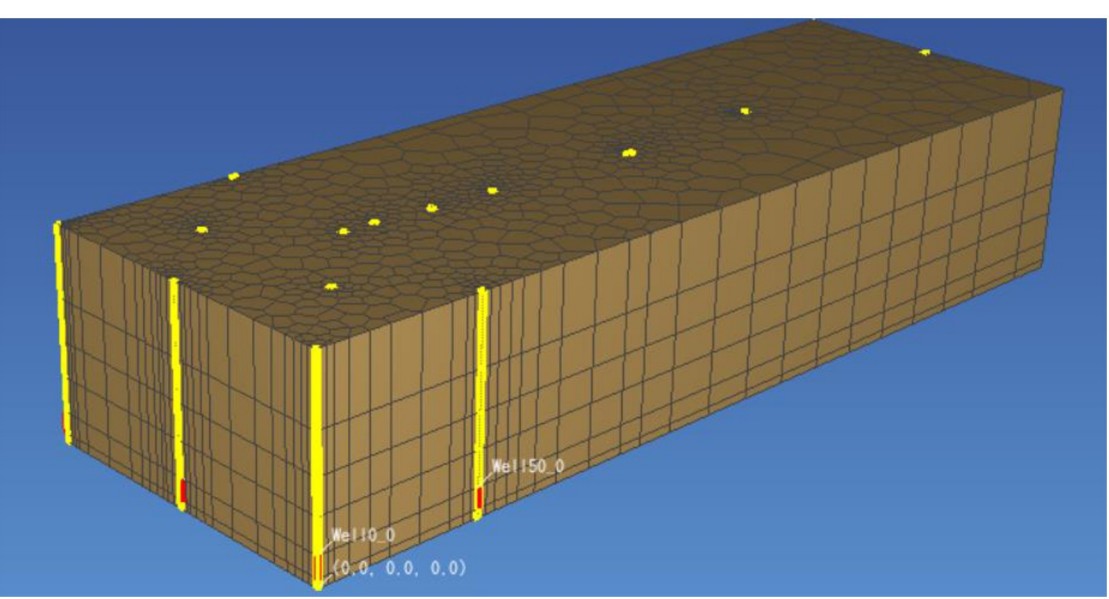

**Fig 5. 3D structure diagram of mesh.**

## 3.4. Model simulation accuracy and verification

We used the linear correlation coefficient to compare the simulated and measured value of pH and $SO_4^{2-}$ concentration. The expression of the linear correlation coefficient is as follows:

$$r = \frac{\sum_{i=1}^{n}\left(C_i - \overline{C_i}\right)\left(b_i - \overline{b_i}\right)}{\sqrt{\sum_{i=1}^{n}\left(C_i - \overline{C_i}\right)^2}\sqrt{\sum_{i=1}^{n}\left(b_i - \overline{b_i}\right)^2}} \tag{14}$$

where, $C_i$ is the simulated concentration, and $\overline{C_i}$ is the mean value of the simulated value; $b_i$ Observed concentration, $\overline{b_i}$ is the mean value of the observed value. Near the simulation area, there are three typical observation holes: W1, W2, W3. Three observation holes were sampled respectively. The pH value is for field test, and $SO_4^{2-}$ were sent to the mining area laboratory for analysis and test.

As shown in Table 9, the calibration involved 9 data points per well. In the calibrate results of three observation wells, the maximum error of pH value is 2.711 and the maximum average error is 0.760. The maximum error of $SO_4^{2-}$ concentration was 0.033mol/kg and the maximum average error is 0.0167mol/Kg. As can be seen from Table 9, correlation coefficients between simulated values and observed values were calculated. The correlation coefficients of the three wells ranged from 0.9433 to 0.9990. The simulation accuracy of this study is above 90%.

**Table 6. Aquifer stratification information.**

| Aquifer number | 1 | 2 | 3 | 4 | 5 | 6 |
|---|---|---|---|---|---|---|
| Top (m) | 60 | 40 | 28.5 | 18.5 | 8.5 | 2.5 |
| Bottom (m) | 40 | 28.5 | 18.5 | 8.5 | 2.5 | 0 |
| Thickness (m) | 20 | 11.5 | 10 | 10 | 6 | 2.5 |

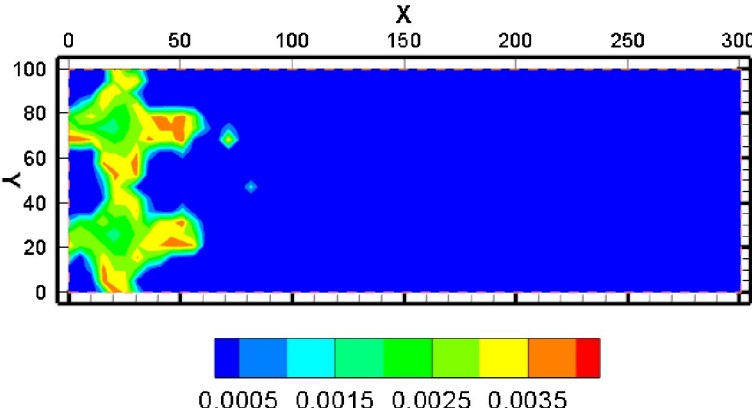

**Fig 6. The concentration distribution of UO22+.**

# 4. Results and discussions

The reactive transport model (RTM) is designed to illustrate the variation of groundwater components and mineral components during uranium leaching.

## 4.1 The distribution of groundwater components

**4.1.1 The concentration distribution of $UO_2^{2+}$.**   The simulation results show that after a year of mining, the concentration of pH in the solution and the increase of the uranium concentration and the increase of the uranium content were reprecipitated. From the distribution of uranium concentration, the uranium concentration was higher near the pumping well (Fig 6). The highest concentration of $UO_2^{2+}$ appeared between the injection well and pumping well. The max concentration of $UO_2^{2+}$ exceeded 0.004mol/kg (Fig 6).

**4.1.2 The concentration distribution of $SO_4^{2-}$.**   In the process of acid leaching, the amount of sulphuric acid is large, so the $SO_4^{2-}$ concentration is an object that needs to be seriously described. In the area of the mining area, the amount of uranium was reduced, and the degree of volume reduction was more than 0.01%. The blue area shows the area where the mineral is dissolved, and the region of the uranium mining is dissolved, and as the solution is moved to the extraction well, the uranium mine is gradually precipitated, but the amount of precipitation is far less than that of the mine, which is far less than the degree of solubility of the liquid well, which can be more than 0.006% but not more than 0.008% (Fig 7). The

**Table 7. The aquifer parameters.**

| Parameters | Value |
|---|---|
| Aquifer thickness(m) | 60 |
| Rock grain density(kg/m$^3$) | 2600 |
| Porosity | 0.085 |
| K: absolute permeability(m$^2$) | $3\times10^{-12}$ |
| Temperature (pe | 9 |
| Rock grain specific heat (J/(Kgl pe | 920 |
| Formation heat conductivity (W/(mperm | 2.51 |
| Pressure (MPa) | 0.1 |
| Salinity (NaCl mass fraction) | Water components |

**Table 8. The mineral reaction kinetic parameters used in the model.**

| Minerals | Chemical formula | Surface area(cm2/g) | Parameters for kinetic rate law | | | | | | | |
|---|---|---|---|---|---|---|---|---|---|---|
| | | | Neutral mechanism | | Acid mechanism | | | Base mechanism | | |
| | | | $k_{25}$(mol/m$^2$/s) | $E_a$ (KJ/mol) | $k_{25}$(mol/m$^2$/s) | $E_a$ (KJ/mol) | N (H$^+$) | $k_{25}$(mol/m$^2$/s) | $E_a$ (KJ/mol) | n(OH$^-$) |
| Calcite | $CaCO_3$ | 9.8 | Equilibrium | | | | | | | |
| Anhydrite | CaSO4 | 9.8 | Equilibrium | | | | | | | |
| Quartz | $SiO_2$ | 9.8 | $1.023\times10^{-14}$ | 87.7 | | | | | | |
| Illite | $K_{0.6}Mg_{0.25}Al_{1.8}(Al_{0.5}Si_{3.5}O_{10})(OH)_2$ | 151.6 | $1.660\times10^{-13}$ | 35.0 | $1.047\times10^{-11}$ | 23.6 | 0.34 | $3.020\times10^{-17}$ | 58.9 | -0.4 |
| K-feldspar | $KAlSi_3O_8$ | 9.8 | $3.890\times10^{-13}$ | 38.0 | $8.710\times10^{-11}$ | 51.7 | 0.5 | $6.310\times10^{-22}$ | 94.1 | -0.823 |
| Chlorite | $Mg_{2.5}Fe_{2.5}Al_2Si_3O_{10}(OH)_8$ | 20.0 | $3.020\times10^{-13}$ | 88.0 | $7.762\times10^{-12}$ | 88.0 | 0.50 | | | |
| Na-smectite | $Na_{0.29}Mg_{0.26}Al_{1.77}Si_{3.97}O_{10}(OH)_2$ | 151.6 | $1.660\times10^{-13}$ | 35.0 | $1.047\times10^{-11}$ | 23.6 | 0.34 | $3.020\times10^{-17}$ | 58.9 | -0.4 |
| Kaolinite | $Al_2Si_2O_5(OH)_4$ | 23.0 | $6.918\times10^{-14}$ | 22.2 | $4.898\times10^{-12}$ | 65.9 | 0.777 | $8.913\times10^{-18}$ | 17.9 | -0.472 |
| Ca-smectite | $Ca_{0.145}Mg_{0.26}Al_{1.77}Si_{3.97}O_{10}(OH)_2$ | 151.6 | $1.660\times10^{-13}$ | 35.0 | $1.047\times10^{-11}$ | 23.6 | 0.34 | $3.020\times10^{-17}$ | 58.9 | -0.4 |
| Gypsum | $CaSO_4$ | 9.8 | 1.6218e-07 | | | | | | | |
| Pyrite | $FeS_2$ | 12.9 | $2.52\times10^{-12}$ | 62.76 | | | | | | |
| Oligoclase | $CaNa_4Al_{1.77}Si_{3.97}O_{10}(OH)_2$ | 10.0 | 1.4454e-13 | 69.80 | | | | | | |
| Hematite | $Fe_2O_3$ | 12.9 | $2.512\times10^{-15}$ | 66.2 | $4.0738\times10^{-10}$ | 66.2 | 1 | | | |
| Muscovite | $KAl_2(AlSi_3O_{10})(OH)_2$ | 152.0 | $3.0200\times10^{-13}$ | 88.0 | $7.7624\times10^{-12}$ | 88.0 | 0.5 | $3.0200\times10^{-13}$ | | |
| Siderite | $FeCO_3$ | 10.0 | $1.26\times10^{-9}$ | 62.76 | $6.46\times10^{-4}$ | 36.1 | 0.5 | | | |
| Dolomite | $CaMg(CO_3)_2$ | 12.9 | $2.52\times10^{-12}$ | 62.76 | $2.34\times10^{-7}$ | 43.54 | 1 | | | |
| Ankerite | $CaMg_{0.3}Fe_{0.7}(CO_3)_2$ | 9.8 | $1.26\times10^{-9}$ | 62.76 | $6.46\times10^{-4}$ | 36.1 | 0.5 | | | |
| Magnesite | $MgCO_3$ | 10.0 | $4.5709\times10^{-10}$ | 23.50 | $4.1687\times10^{-07}$ | 14.4 | 1 | | | |

simulation results show that after a year of mining, the concentration of pH in the solution and the increase of the uranium concentration and the increase of the uranium content were reprecipitated. From the distribution program of uranium concentration, the uranium concentration was low, and the uranium concentration was high near the pumping well.

**4.1.3 The concentration distribution of H$^+$.** A significant component affecting ISL is the H$^+$ concentration. Fig 8 illustrates the pH shift after a year of mining. The pH value is relatively low in the mining area's interior. The pH value gradually increased as the distance from the mining region grew. Fig 9 also shows that there won't be any more noticeable alterations if the lower pH range is essentially extended to 50 meters. The migration range of H$^+$ is not as far as that of other components. The reason may be attributed to the active hydrogeochemical properties of H$^+$, which need to participate in many chemical re-actions during the migration.

**Table 9. The difference between calculated values and observed values at observation well.**

| Observed well | Observed point | Index | Max error | Average error | Average relativity error | Correlation coefficient |
|---|---|---|---|---|---|---|
| W1 | 9 | pH | 2.711 | 0.7600 | 0.122 | 0.9433 |
| W1 | 9 | $SO_4^{2-}$ | 0.033 | 0.0167 | 0.107 | 0.9990 |
| W2 | 9 | pH | 0.3 | 0.0640 | 0.033 | 0.9905 |
| W2 | 9 | $SO_4^{2-}$ | 0.032 | 0.0166 | 0.097 | 0.9918 |
| W3 | 9 | pH | 0.605 | 0.1490 | 0.088 | 0.9702 |
| W3 | 9 | $SO_4^{2-}$ | 0.01 | 0.0020 | 0.020 | 0.9989 |

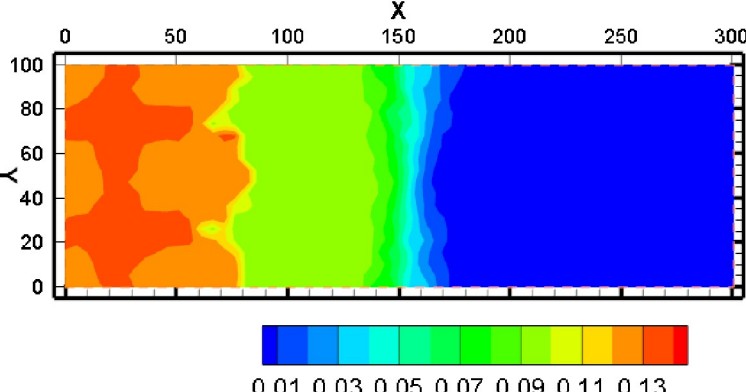

**Fig 7. The concentration distribution of SO42-.**

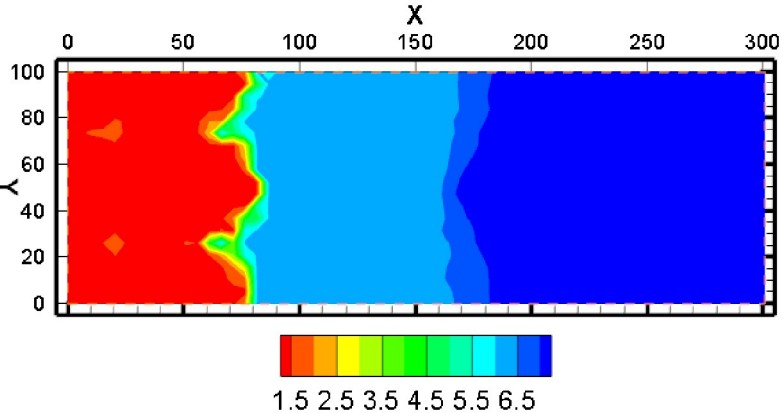

**Fig 8. The distribution of pH value.**

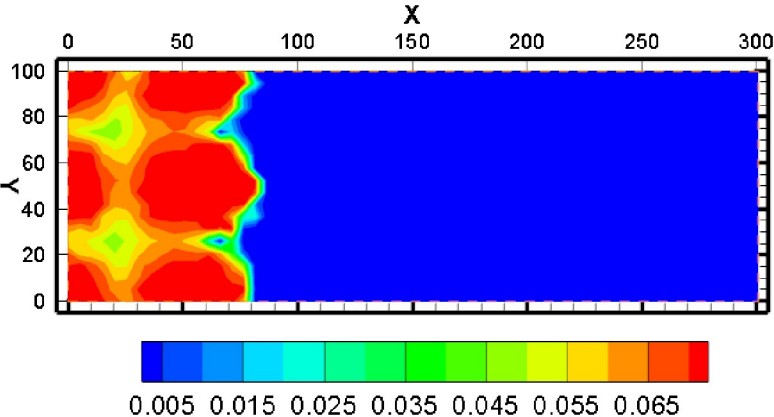

**Fig 9. The concentration distribution of H+.**

**4.1.4 The concentration distribution of $K^+$, $Na^+$, $Ca^{2+}$, $Mg^{2+}$, $HCO_3^-$, $AlO_2^-$, $Cl^-$.** $Ca^{2+}$ concentration varies from 0.01 mol/kg to 0.012 mol/kg after one year. The concentration of $Ca^{2+}$ steadily decreased from high to low and may exceed 0.012mol/Kg as it migrated away from the mining region. This occurs because of the diffusion and outward migration of $H^+$, which causes the dissolving of carbonate minerals outside the mining region. The $Ca^{2+}$ content rapidly decreases from 0.012 mol/kg to the background level after reaching a high at 150–180 m from the mine area. The progressive dissolution of magnesium carbonate minerals causes the concentration of $Mg^{2+}$ to gradually grow and can surpass 0.018mol/Kg during migration of $Mg^{2+}$ to the outside of the mining area. About 150 meters away from the mining location, a front of $Mg^{2+}$ concentration appears. Then, the concentration of $Mg^{2+}$ dropped from 0.018mol/Kg to background levels quickly.

$Na^+$ and $K^+$ have similar migration trend. The greatest concentration of $Na^+$ might exceed 0.065mol/Kg after one year of mining, and the concentration of $Na^+$ steadily fell as it migrated to the mining area's edges. At 150 meters from the mine, there is a concentration front of $Na^+$ where the concentration can quickly drop from more than 0.065 mol/kg to background levels. As the $K^+$ concentration migrates away from the mining area, the concentration gradually decreases, with the greatest concentration exceeding 0.01 mol/kg. At 150m from the mine, there is a concentration front of $K^+$ where the concentration can quickly drop from more than 0.065mol/Kg to background levels. Because feldspar and clay minerals out of the mining area will gradually dissolve with the outward migration of $H^+$, the concentration distribution of $Na^+$ and $K^+$ exhibits this pattern.

Due to the influence of pH value, the concentration of $HCO_3^-$ in the mining area is the lowest. With the increasing distance from the mining area, the concentration of $HCO_3^-$ firstly increased and then decreased. This is because $HCO_3^-$ and $H^+$ interact significantly. Although there is a large amount of $CO_3^{2-}$ in highly acidic regions, it is difficult to generate $HCO_3^-$ due to low pH. In the area far from the mining area, the groundwater has sufficient carbonate dissolution capacity and low concentration of $H^+$, the above distribution pattern appears.

The concentration of $AlO_2^-$ in the mining area is relatively low, and the concentration front is about 120m away from the mining area. Since $AlO_2^-$ is unstable in both acidic and alkaline environments, the concentration of $AlO_2^-$ is higher in the weakly acidic environment, and lower near the mining area and far away from the mining area.

In Fig 10, the concentration of $Cl^-$ keeps stabilization relatively. The relatively low concentration of $Cl^-$ in the mine area is because the concentration of $Cl^-$ in the injected solution is slightly less than the groundwater background value. Chlorine-containing minerals generally have a strong solubility and are usually dissolved, therefore, the concentration of $Cl^-$ can only be diluted by an injected solution.

## 4.2 Mineral volume fraction variation, dissolution, and precipitation

By means of numerical simulation of mining situation in the mining area, the change of the migration situation of the chemical component of groundwater and the composition of each mineral component in groundwater were obtained, and the change of the pore permeability structure was obtained, and the simulation results can be observed to analyze the trend and influence factors of them.

**4.2.1 Mineral volume fraction changes at observed well.** Through the analysis of the simulation results of the change in the volume fraction of the mineral species at the observation point, the dissolution and precipitation state of minerals can be clearly identified. A positive change in the volume fraction of the mineral indicates that the mineral has precipitated, and a negative change in the volume fraction of the mineral indicates that the mineral has

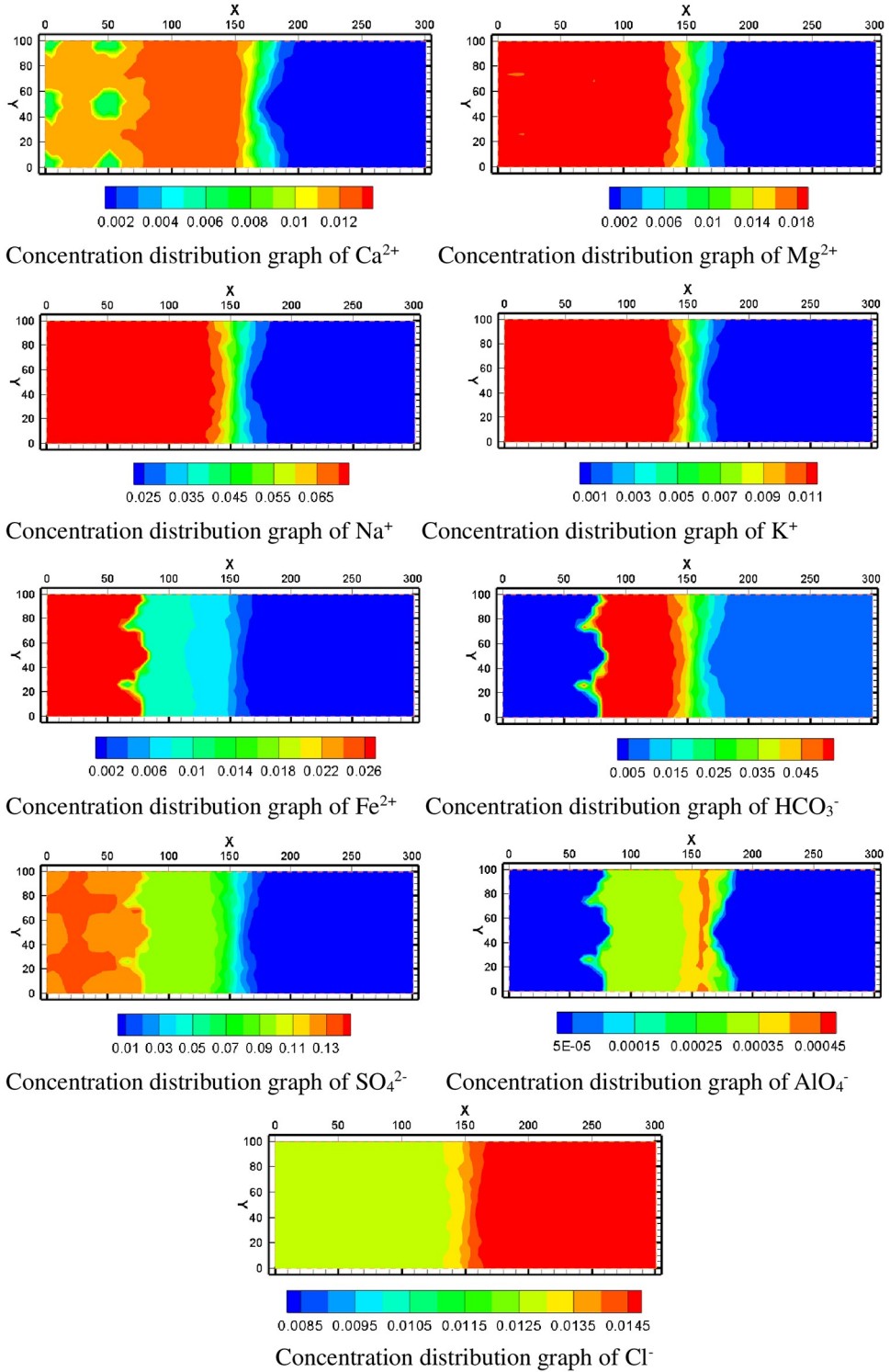

Concentration distribution graph of $Ca^{2+}$ · Concentration distribution graph of $Mg^{2+}$

Concentration distribution graph of $Na^+$ · Concentration distribution graph of $K^+$

Concentration distribution graph of $Fe^{2+}$ · Concentration distribution graph of $HCO_3^-$

Concentration distribution graph of $SO_4^{2-}$ · Concentration distribution graph of $AlO_4^-$

Concentration distribution graph of $Cl^-$

**Fig 10. Concentration distribution graph of selected parameters.**

dissolved. To observe the dissolution and precipitation of various minerals, the simulated values of the volume fraction changes of several mineral components in the observation well W1 nearest to the mining area were analyzed.

Calcite is a strongly dissolved mineral. After mining, the calcite begins to dissolve, and the dissolution rate increases gradually. The volume fraction of calcite decreases the most about 0.1 years after mining. The actual simulation results show that the calcite has been completely dissolved at the observation point.

According to the simulation results of well W1, anhydrite is a precipitated mineral. After mining, anhydrite began to precipitate, and the precipitation rate gradually increased. About 0.1 years after mining, the precipitation reaches its peak, and then the volume integral of anhydrite decreases. About 0.8 years later, the volume fraction of anhydrite is reduced to its initial value. Anhydrite precipitates and then dissolves completely until about 0.8 years after mining. The volume fraction of gypsum continues to increase after mining. This trend slows down rapidly about 0.8 years after production. The contrast between the dissolution and precipitation of gypsum and anhydrite indicates that a large part of the precipitation of gypsum comes from the redissolution of anhydrite. When the anhydrite is completely dissolved, the gypsum also stops precipitation.

The volume fraction of uranium ore increases first and then decreases. The variation of uranium volume fraction shows a wave pattern, reaching a peak at 0.1 years after mining. The volume fraction increases by as much as 0.008. This is because the leaching solution in the mine contains a high concentration of uranium in the process of moving out. Therefore, with the consumption of $H^+$, uranium ore is precipitated again. It is also noted that at the observation point, the precipitated uranium ore begins to dissolve after all the calcite is consumed. Thus, due to the presence of calcite, the acid solution dissolves the calcite first, buffering the dissolution of the uranium ore. Hematite began to precipitate after the ISL was started. As the lixiviant moved outward, hematite gradually precipitated. However, the precipitation trend of hematite also decreased sharply in 0.1 year. This is because the calcite minerals in the observation point are consumed at this time, hence the pH value at the observation point dropped sharply, and the water environment at the observation point also weakened the precipitation trend of hematite minerals.

K-feldspar minerals also begin to dissolve after 0.1 years of mining. This indicates that a large amount of dissolution of K-feldspar occurred after the dissolution of calcite.. Oligoclase and pyrite were also affected by the dissolution of calcite, and these two minerals began to dissolve significantly at 0.1 year.

In clay minerals, illite and chlorite tend to precipitate first and then dissolve. Calcium montmorillonite and sodium montmorillonite continue to dissolve. Kaolinite continues to precipitate. Their dissolution and precipitation change abruptly at 0.1 years after mining. This indicates that the dissolution and precipitation of these five minerals are also related to calcite.

Siderite and ankerite were precipitated first and then dissolved, which was also due to the consumption of calcite at the observation point in 0.1 year, so that the two minerals were rapidly dissolved. Muscovite mineral is similar to hematite, which was precipitated first and then unchanged.

Through the analysis of the dissolution and precipitation state of these minerals, at the observation point, the reaction of mineral composition existed in the order. Calcite reacted first and became the source of anhydrite and gypsum minerals, while other minerals reacted further after calcite reaction (Fig 11).

**4.2.2 Dissolution and precipitation of minerals in simulation area.** When the acid mining fluid was injected into the aquifer, the uranium mineral was dissolved in the groundwater, and this was the major mechanism of the ISL. In Fig 12, the blue area means uranium

**Fig 11. The mineral volume fraction changes at observed W1.**

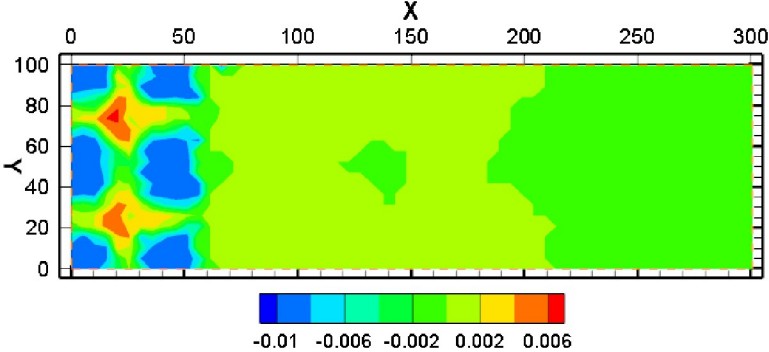

**Fig 12. The volume fraction change of uranium.**

dissolution, and the red area means uranium precipitation. The maximum increase in the volume fraction of uranium ore can exceed 0.006% but not exceeded 0.008%. The volume fraction of uranium ore can be reduced by more than 0.01%. From the simulation results, the uranium ore was mainly dissolved near the injection well, which was the main source of uranium element in the mining area and its nearby liquid phase. The volume fraction of uranium was reduced by 0.01% near the injection hole. The volume fraction of uranium can increase to 0.006% near the pumping well. This is because along the flow field flow line direction, the acid in the leaching solution is gradually consumed. As the pH increases, the dissolved uranium is precipitated again. In the outer area of the mining area, the content of uranium element is relatively low.

Calcite is one of the main dissolved minerals in the study area and one of the sources of calcium ions in the mining area. It can be seen from the calcite volume fraction change chart that the calcite is in the dissolved state in the mining area, and the volume fraction can decrease by 2.8%. When the leaching solution is injected into the aquifer, it causes the dissolution of calcite. Anhydrite and gypsum precipitate due to high concentrations of $Ca^{2+}$ and $SO_4^{2-}$. As can be seen from the spatial distribution of volume fraction changes of anhydrite and gypsum, their mineral volume fraction increases less near injection Wells. The increase in volume fraction is larger in areas far from the injection well. This is because the pH near the water injection well is low and the groundwater environment near the water injection well is in a strongly acidic state. Therefore, although there are high concentrations of $Ca^{2+}$ and $SO_4^{2-}$, precipitation is not easy to occur. In Fig 13, near the injection well, the increase in the volume fraction of anhydrite is less than 0.5%, and the increase in the volume fraction of gypsum is about $6.5 \times 10^{-7}$. Near pumping well, the maximum increase of anhydrite volume fraction is greater than 3.5%, and the maximum increase of gypsum volume fraction is greater than $9 \times 10^{-7}$.

K-feldspar and oligoclase are two feldspar minerals, and their volume fractions are like each other. But K-feldspar has a greater range of volume fractions changes. Near the injection well,

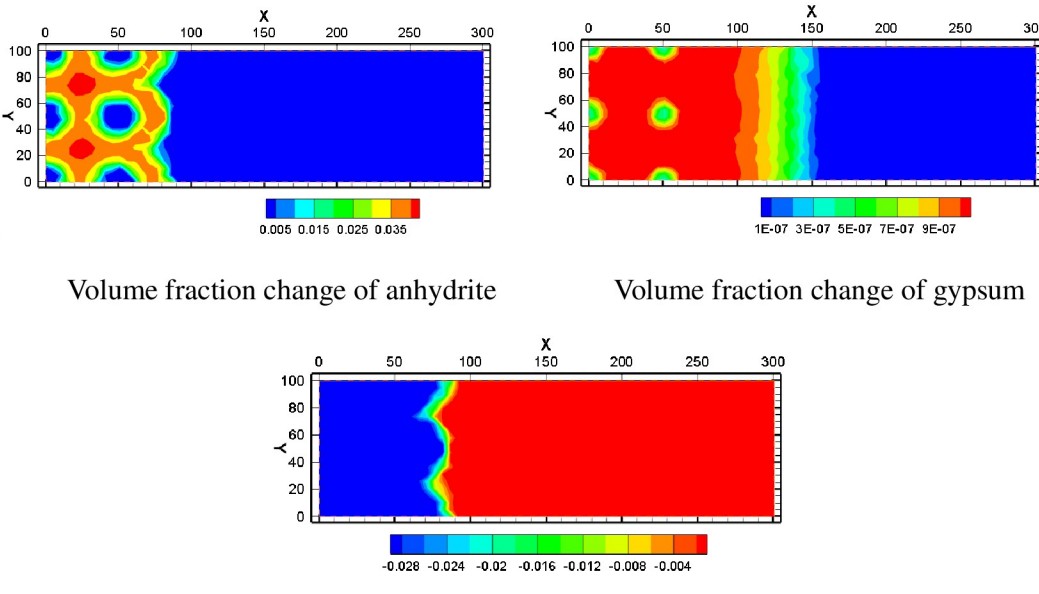

Volume fraction change of anhydrite          Volume fraction change of gypsum

Volume fraction change of calcite

**Fig 13. Volume fraction change of anhydrite, gypsum and calcite.**

the volume fraction of feldspar mineral decreases obviously. The volume fraction of K-feldspar decreased by $-9 \times 10^{-6}$, and the volume fraction of oligoclase decreased by $-4.5 \times 10^{-7}$. The decline trend of feldspar mineral volume fraction gradually decreases with the distance from injection wells. There is little variation in the volume fraction of quartz minerals. Hematite and pyrite are two kinds of iron bearing minerals in mining area. The hematite in the mining area is generally in the state of precipitation, while the pyrite is in the state of dissolution. The high content of iron in mining area is closely related to the dissolution and precipitation of iron-bearing minerals. The variation of their volume fractions is shown in Fig 14.

The simulation included five clay minerals: Ca-smectite, Na-smectite, illite, chlorite and kaolinite. In the process of ISL, Ca-smectite and Na-smectite are both in a dissolved state. In Fig 15, the volume fraction of Na-smectite decreases more than that of Ca-smectite. Illite and chlorite may precipitate or dissolve depending on the location of different spatial regions. They dissolve in the mining area near where the acid solution is injected, and as they move away from the mining area, they become precipitates. Among them, the amount of illite precipitation is larger. Kaolinite, however, remains in a state of precipitation. This is because the source of kaolinite is relatively rich, feldspar minerals and clay minerals can form kaolinite after dissolution.

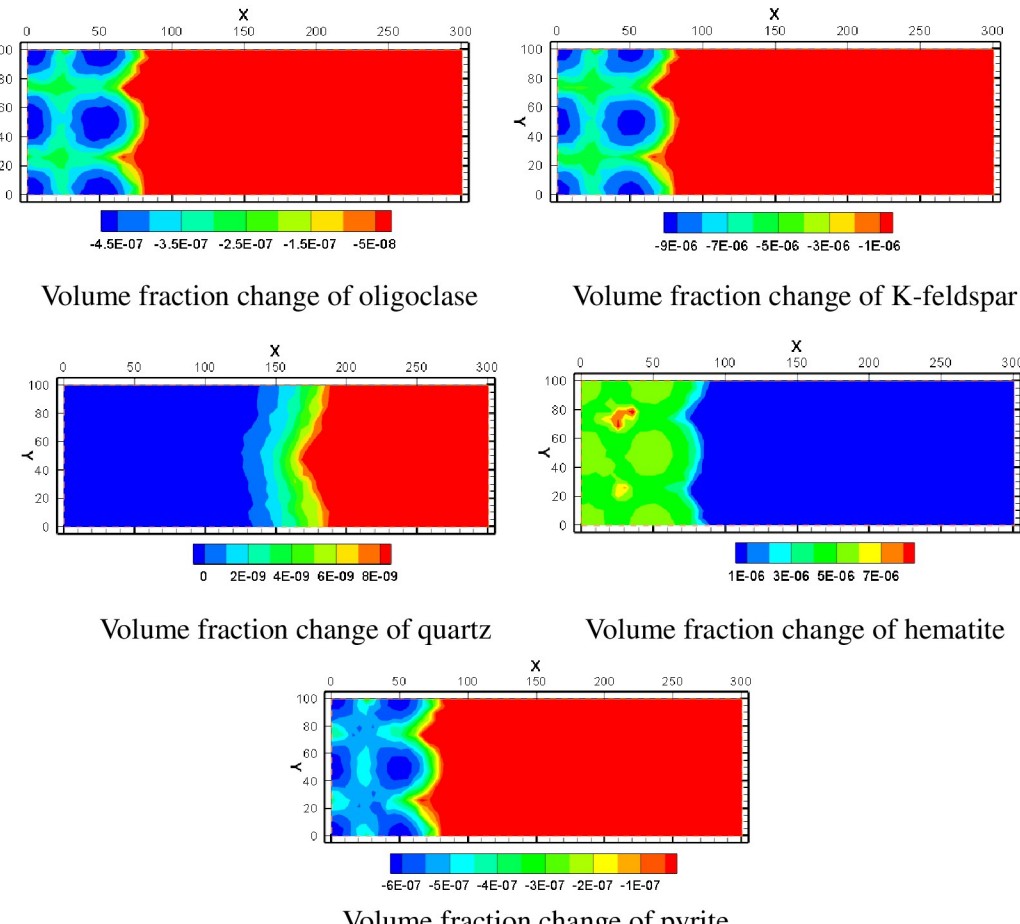

**Fig 14. Volume fraction change of various minerals.**

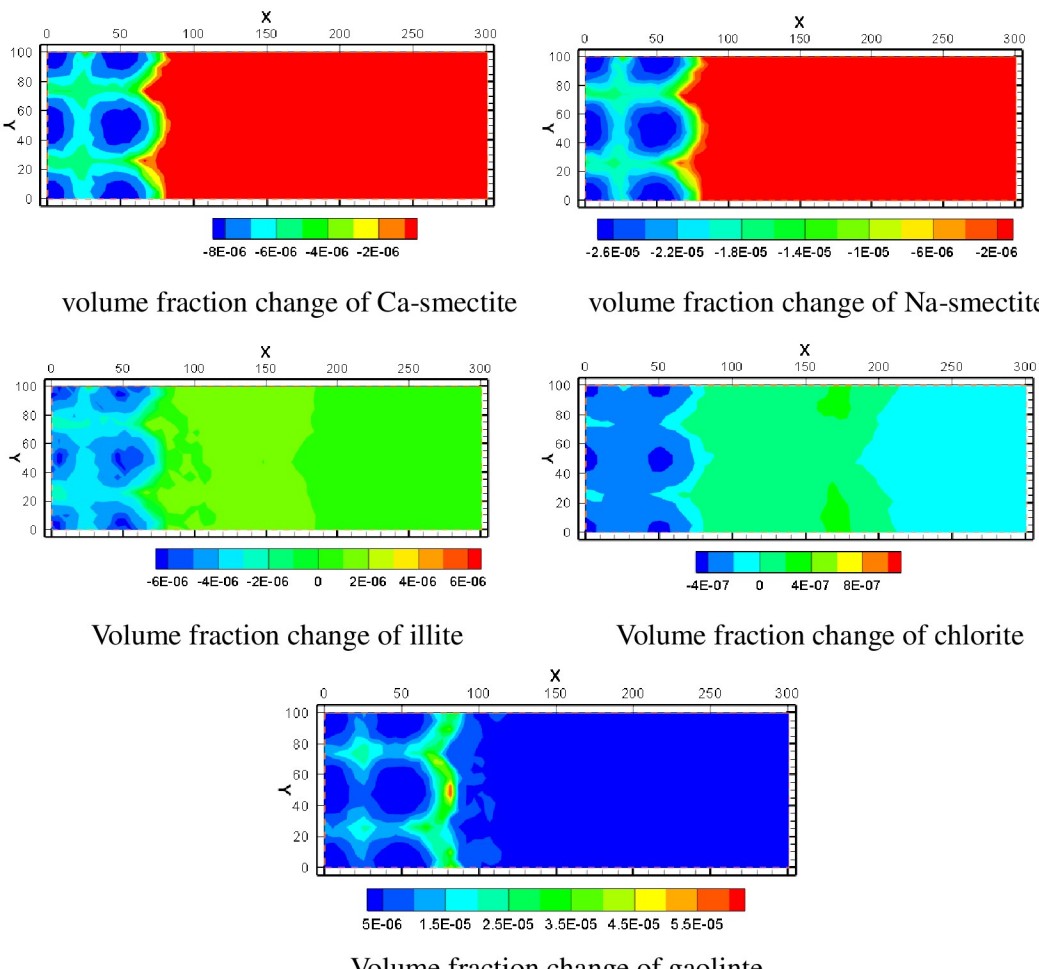

**Fig 15. Volume fraction change of various minerals.**

Muscovite, siderite and ankerite were added to the simulation as secondary minerals. According to the simulation results, muscovite minerals mainly appear in the center of the water injection well and in the area within a certain distance between the pumping well and the water injection well. In the middle of the percolation zone between pumping well and injection well, the volume integral increases little. The volume fractions of siderite and ankerite are very similar. According to the volume fraction change chart, the two kinds of minerals mainly precipitate outside the mining area, and the precipitation trend gradually decreases from the edge of the mining area to the outside of the mining area. It shows that these two minerals are greatly affected by the lixiviant transferred from the mining area. In the mine, siderite and ankerite are difficult to precipitate because of strong acidity. When the solution in the mining area migrates and diffuses outwards, the pH value gradually decreases and the saturation index of many minerals is large, precipitating occurs immediately. With the increase of migration distance, each ion gradually consumes, and the precipitation trend decreases. The spatial distribution of their volume fraction changes is shown in Fig 16.

**4.2.3 Change of porosity and permeability.** Porosity and permeability is the important parameters of the reservoir. Izgec explored the porosity with significant high in rock

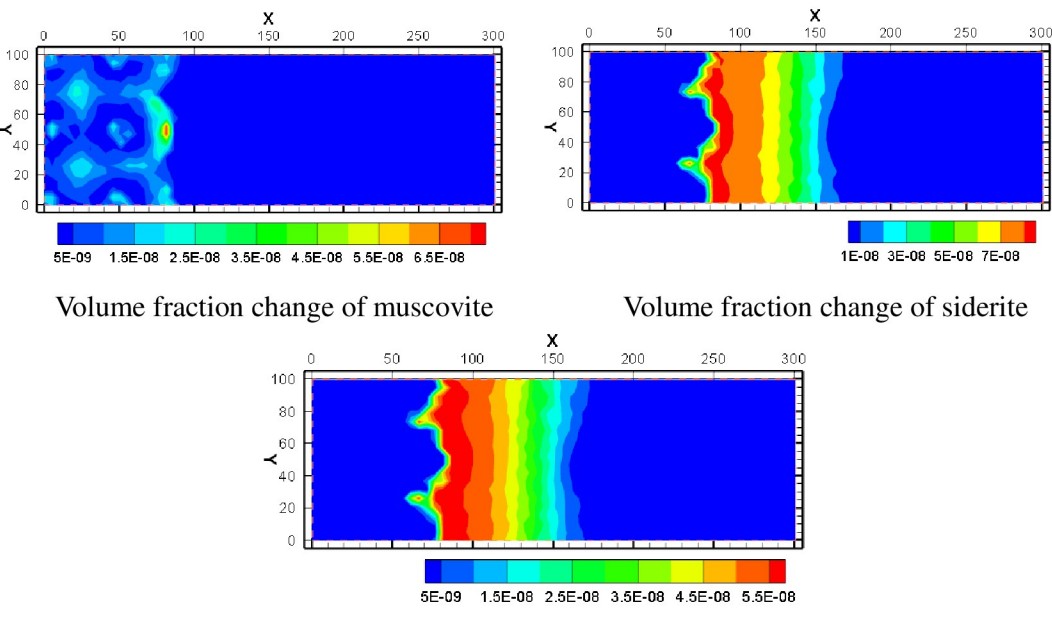

Fig 16. Volume fraction change of various minerals.

permeability using classical porosity-permeability models [38], and granite showed different behavior in dependent of permeability and porosity [39]. The changing trend of porosity and reservoir permeability was relatively similar as shown in Fig 17. The porosity increases to 0.12 near the injection well. This is because the acid concentration near the injection well is high, and the saturation index of minerals is relatively low, hence the minerals dissolution obvious near the injection well. The porosity near the pumping well can be reduced to 0.07. Although the pH near the pumping well is still acidic, it is less acidic than in the injection well, and some minerals (such as gypsum) are gradually precipitated due to a large increase in the concentration of various ions in the groundwater. The permeability increases near the injection well and decreases near the pumping well. The maximum permeability near the injection well is slightly greater than $8\times10^{-12}$. The minimum permeability near the pumping well is slightly less than $2\times10^{-12}$.

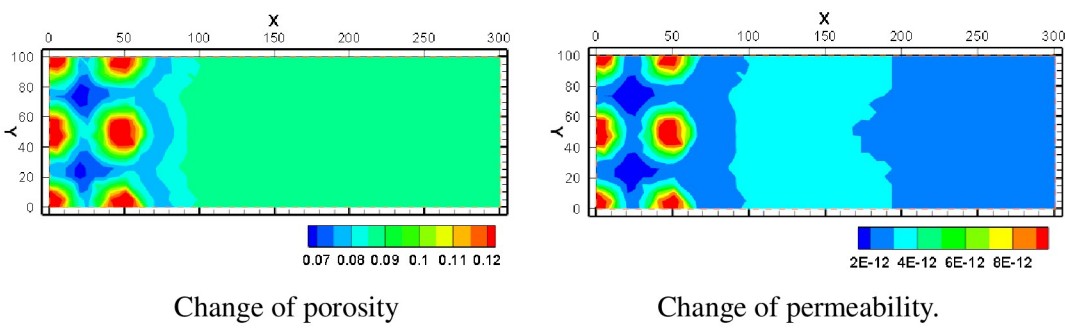

Fig 17. Change of porosity and permeability.

## 5. Conclusions

In this study, the reactive transport model of ISL considers the interaction of various minerals, various hydrogeochemical components and the changes in porosity and permeability properties simultaneously. The model depicts the migration distance and change trend of uranium elements, $H^+$ and various hydro chemical components that may be involved in mining. At the same time, the model describes the changing trend of uraninite, feldspar minerals, clay minerals. The state of minerals that dissolution or precipitation is described. And the effect of ISL on the porosity and permeability has been revealed. The main conclusions are as follow:

1. The reaction property of $H^+$ is active, and the injection amount is also large. However, because it will react with various minerals in the mining area, its influence distance will not be too far, and the influence distance of $K^+$, $Na^+$, $Ca^{2+}$, $Mg^{2+}$ and other major metal cations is relatively far with the development of the mining process Their concentration front is about 150m away from the outside of the mining area. At the same time, the dissolution area of minerals mainly occurs inside the mining area, and its approximate range is close to the coverage range of $H^+$. This is because H+ is chemically active and reacts quickly with various minerals. Only a small amount of $H^+$ migrated outside the mine area.

2. Calcite is the most easily dissolved mineral, and other minerals will have significant dissolution after the dissolution of calcite. Calcite is also one of the sources of gypsum precipitation. Due to the consumption of calcite, the leaching solution will gradually dissolve uranium ore, hematite, and other minerals. The uranium is basically limited in the mining area, and the trend of outward migration is very weak.

3. Uraninite is mainly dissolved near the injection well, but some uranium elements will precipitate out again with the migration to the pumping well. The main dissolved minerals include K-feldspar, oligoclase, pyrite, calcite, Na-smectite, and Ca-smectite. These minerals provide the main source of metal cations in the underground water of the mining area. Illite and chlorite dissolve inside the mining area and precipitate outside the mining area. Hematite as precipitation minerals, mainly precipitate inside the mining area. Siderite and ankerite are precipitated outside the mining area as precipitation minerals. Gypsum minerals are precipitated inside and outside the mining area.

4. ISL will cause changes of porosity and permeability of the mining area. Near the injection well, many minerals in the aquifer will dissolve, resulting in increased porosity and permeability. Near the pumping well, the mineral precipitation will lead to the reduction of porosity and permeability, which may cause the blockage of the pore channel, and will have a negative impact on the mining efficiency.

During the mining process, the scope of strong acidity of groundwater is principally in the mining area. The acidic substances overflowing to the periphery of the mining area are neutralized by minerals, resulting in a rise of numerous elements. As well as causing changes in porosity and permeability.

## Acknowledgments

The authors would like to thank Mr. Yiwei Li and Prof. Jili Wen from Beijing Research Institute of Chemical Engineering and Metallurgy for advising the work. Editors and two anonymous reviewers are also thanked for the valuable comments in improving the quality of the manuscript.

## Author Contributions

**Conceptualization:** Akhtar Malik Muhammad, Zhonghua Tang.

**Data curation:** Haibo Li, Zhonghua Tang.

**Formal analysis:** Haibo Li.

**Software:** Haibo Li.

**Supervision:** Zhonghua Tang.

**Validation:** Akhtar Malik Muhammad, Zhonghua Tang.

**Visualization:** Zhonghua Tang.

**Writing – original draft:** Akhtar Malik Muhammad.

**Writing – review & editing:** Akhtar Malik Muhammad.

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
