## [Decision Letter · Decision Letter 0]

11 May 2023

PONE-D-23-09655Assessment of groundwater quality with reference to minerals in situ leaching uranium in Bayanwula mining area, ChinaPLOS ONE

Dear Dr. Akhtar,

Thank you for submitting your manuscript to PLOS ONE. After careful consideration, we feel that it has merit but does not fully meet PLOS ONE’s publication criteria as it currently stands. Therefore, we invite you to submit a revised version of the manuscript that addresses the points raised during the review process.

We look forward to receiving your revised manuscript.

Kind regards,

Vasanthavigar Murugesan, M.Sc., Ph.D.,https://orcid.org/my-orcid?orcid=0000

Academic Editor

PLOS ONE

Journal Requirements:

"No Funder, Fee will be paid by authors"

"Authors with competing interests" 

7. We note that Figure 1 in your submission contain [map/satellite] images which may be copyrighted. All PLOS content is published under the Creative Commons Attribution License (CC BY 4.0), which means that the manuscript, images, and Supporting Information files will be freely available online, and any third party is permitted to access, download, copy, distribute, and use these materials in any way, even commercially, with proper attribution. For these reasons, we cannot publish previously copyrighted maps or satellite images created using proprietary data, such as Google software (Google Maps, Street View, and Earth). For more information, see our copyright guidelines: http://journals.plos.org/plosone/s/licenses-and-copyright.

8. Please clarify the Table 1 "Table 1. Well flow rate." in page "7" and Table 1 "Table 1 Secondary mineral in simulation" in page s"10 and 11" .

Additional Editor Comments:

This manuscript need to improve the quality. Hence this ms need major revision

Reviewers' comments:

Reviewer's Responses to Questions

**Comments to the Author**

1. Is the manuscript technically sound, and do the data support the conclusions?

Reviewer #1: Partly

Reviewer #2: No

2. Has the statistical analysis been performed appropriately and rigorously? 

Reviewer #1: Yes

Reviewer #2: No

3. Have the authors made all data underlying the findings in their manuscript fully available?

Reviewer #1: No

Reviewer #2: No

4. Is the manuscript presented in an intelligible fashion and written in standard English?

Reviewer #1: No

Reviewer #2: No

5. Review Comments to the Author

Reviewer #1: The authors have carried out a study on "Assessment of groundwater quality with reference to minerals in situ leaching uranium in Bayanwula mining area, China". However, after reviewing this manuscript, there are the following problems:

1.Abstract: Please refine the main research highlights to improve readability, otherwise it will reduce the quality of your manuscript. The current summary is too long.

2.Introduction: It is necessary to reorganize the structure and sort out the current research status. The current introduction structure cannot effectively guide readers in reading the manuscript. I have annotated the details in the attachment.

3.In section of 3.4 Initial parameters of the model, there are too many chapters and each paragraph has very little content. It is recommended to integrate them.

4.What is the basis for the author's determination of minerals in the manuscript?

5.3.4.5 Model simulation accuracy and verification, There are many writing errors.

6.4. Results and discussions, it is just an explanation of the results, lacking in-depth discussion. In addition, the entire writing process is a bit chaotic and difficult to understand.

7.Conclusions: 1.: Please refine the main research highlights to improve readability, otherwise it will reduce the quality of your manuscript.

In addition, there are two important issues:

(1)Suggest adding line numbers and page numbers, otherwise it will increase the difficulty of the review.

(2)There are too many writing errors and syntax error in the manuscript, and the manuscript language needs to be improved by professionals or institutions, otherwise it is not suitable for publication.

Reviewer #2: 1.Introduction part is illogical. The reason for the research has not been clarified. What is the novelty in the manuscript? Methodology or study area?

2. Geological map and section are absent.

3. water-bearing rock should be replaced by aquifer.

4. Uranium orebody is not mentioned in Study area.

5. 40~60m is not correct writing in English.

6. Groundwater flow direction is missing in Hydrogeological conditions.

7. The English should be polished. Too many very long sentences make the manuscript very hard to be understood.

8. The results of numerical simulation are not robust. More evidence is needed to prove it.

6. PLOS authors have the option to publish the peer review history of their article (what does this mean?). If published, this will include your full peer review and any attached files.

Reviewer #1: No

Reviewer #2: No

---

## [Author Response · Author response to Decision Letter 0]

25 Nov 2023

Reviewer Comments

Comments Reviewer #1

The authors have carried out a study on "Assessment of groundwater quality with reference to minerals in situ leaching uranium in Bayanwula mining area, China". However, after reviewing this manuscript, there are the following problems:

1.Abstract: Please refine the main research highlights to improve readability, otherwise it will reduce the quality of your manuscript. The current summary is too long.

RESPONSE： The authors are thankful to the reviewer for the helpful comments on improving our manuscript. we have revised the entire document to contextualize our findings and provide a more elaborate understanding of this research work and their implications as per the reviewer's suggestion. The entire document has been rearranged and rewritten. 

We have revised the abstract. The main advantages of this study have been included in the abstract and introduction. ( line 12 to 32, line 130 to 152)

2. Introduction: It is necessary to reorganize the structure and sort out the current research status. The current introduction structure cannot effectively guide readers in reading the manuscript. I have annotated the details in the attachment. 

RESPONSE： We have further summarized the current research status and reorganized the introduction as per the reviewer's suggestion. (line 40 to 152).

3. In section of 3.4 Initial parameters of the model, there are too many chapters and each paragraph has very little content. It is recommended to integrate them. 

RESPONSE： We have updated the chapters and rearranged them as per the reviewer's suggestion. (Chapter’3.2 mathematical model).

4.What is the basis for the author's determination of minerals in the manuscript?

RESPONSE： Thanks for your comments. This section is included in this manuscript to give the actual site field understanding.

5.3.4.5 Model simulation accuracy and verification, There are many writing errors.

RESPONSE: We have updated the model accuracy and verification as per the reviewer's suggestion. (Line 412 to 434)

6.4. Results and discussions, it is just an explanation of the results, lacking in-depth discussion. In addition, the entire writing process is a bit chaotic and difficult to understand.

RESPONSE： Thanks a lot. We adjusted it according to two parts: fluid migration and mineral change. We add to the discussion of possible mineral transformation relationships as per the reviewer's suggestion. 

7.Conclusions: 1.: Please refine the main research highlights to improve readability, otherwise it will reduce the quality of your manuscript.

RESPONSE：We rearranged the main research highlights of this research as per the reviewer's suggestion.(line 670 to 672)

In addition, there are two important issues:

(1)Suggest adding line numbers and page numbers, otherwise it will increase the difficulty of the review.

RESPONSE: The line numbers and page numbers has been added in revised manscript.

(2)There are too many writing errors and syntax error in the manuscript, and the manuscript language needs to be improved by professionals or institutions, otherwise it is not suitable for publication.

RESPONSE： Modified as per instructions of entire paper 

Comments Reviewer #2

1.Introduction part is illogical. The reason for the research has not been clarified. What is the novelty in the manuscript? Methodology or study area?

RESPONSE： The authors are grateful for the reviewer’s critical comments regarding the manuscript’s improvement. Introduction part has been modified and rearranged as per reviewer’s guideline and comments. We addressed all your comments and modified our manuscript. We compare the innovation of this study with the gap of previous research in the introduction part.

2. Geological map and section are absent.

RESPONSE： The stratum structure of the study area is relatively flat; we showed the geological structure of the study area in a vertical sequence diagram in Figure 2 and Figure 3.

3. water-bearing rock should be replaced by aquifer.

RESPONSE： Thanks a lot. We replaced the “water-bearing rock” with “aquifer”.

4. Uranium orebody is not mentioned in Study area.

RESPONSE： The uranium orebody has been included and mentioned in the legend ”Sandstone uranium deposits” in Figure-1.

5. 40~60m is not correct writing in English.

RESPONSE： Modified in line 167 to 190 as per reviewer’s guideline and comments. 

6. Groundwater flow direction is missing in Hydrogeological conditions.

RESPONSE： Thanks for your comments. The study area is in a flat high plain with week background groundwater flow. The main impact comes from manual extraction. We have included the groundwater flow direction in Figure 4.

7. The English should be polished. Too many very long sentences make the manuscript very hard to be understood.

RESPONSE： Modified as per instructions of entire paper

8. The results of numerical simulation are not robust. More evidence is needed to prove it.

RESPONSE： We have revised the results and included model accuracy and verification in line number 412 to 434 as per reviewer’s guideline and comments.

---

## [Decision Letter · Decision Letter 1]

27 Feb 2024

PONE-D-23-09655R1Assessment of groundwater quality with reference to minerals in situ leaching uranium in Bayanwula mining area, ChinaPLOS ONE

Dear Dr. Akhtar,

Thank you for submitting your manuscript to PLOS ONE. After careful consideration, we feel that it has merit but does not fully meet PLOS ONE’s publication criteria as it currently stands. Therefore, we invite you to submit a revised version of the manuscript that addresses the points raised during the review process.

We look forward to receiving your revised manuscript.

Kind regards,

Jin Wu

Academic Editor

PLOS ONE

Journal Requirements:

Reviewers' comments:

Reviewer's Responses to Questions

**Comments to the Author**

1. If the authors have adequately addressed your comments raised in a previous round of review and you feel that this manuscript is now acceptable for publication, you may indicate that here to bypass the “Comments to the Author” section, enter your conflict of interest statement in the “Confidential to Editor” section, and submit your "Accept" recommendation.

Reviewer #1: All comments have been addressed

2. Is the manuscript technically sound, and do the data support the conclusions?

Reviewer #1: Yes

3. Has the statistical analysis been performed appropriately and rigorously? 

Reviewer #1: Yes

4. Have the authors made all data underlying the findings in their manuscript fully available?

Reviewer #1: Yes

5. Is the manuscript presented in an intelligible fashion and written in standard English?

Reviewer #1: Yes

6. Review Comments to the Author

Reviewer #1: 1.The research results are too long and need to be condensed into highlights.

2.Professional English users are advised to revise the wording of the manuscript.

7. PLOS authors have the option to publish the peer review history of their article (what does this mean?). If published, this will include your full peer review and any attached files.

Reviewer #1: No

---

## [Author Response · Author response to Decision Letter 1]

22 Apr 2024

Response to review comments

RESPONSE:

The authors are grateful for your technical comments regarding our work. We appreciate your informative and valuable comments. All the references are verified and updated as per suggestions. For your convenience, the updated and corrected references' text color has been altered. 

Reviewer #1: 1. The research results are too long and need to be condensed into highlights.

2. Professional English users are advised to revise the wording of the manuscript.

RESPONSE:

The authors are thankful to the reviewer for the helpful comments on improving our manuscript. we have revised the entire document to contextualize our findings and provide a more elaborate understanding of this research work and their implications as per the reviewer's suggestion. The result part is rearranged, and the English has been improved of the entire document.

---

## [Editor Report · Decision Letter 2]

30 Apr 2024

Assessment of groundwater quality with reference to minerals in situ leaching uranium in Bayanwula mining area, China

PONE-D-23-09655R2

Dear Dr. Akhtar,

We’re pleased to inform you that your manuscript has been judged scientifically suitable for publication and will be formally accepted for publication once it meets all outstanding technical requirements.

Kind regards,

Jin Wu

Academic Editor

PLOS ONE
---

## [Editor Report · Acceptance letter]

3 Jun 2024

PONE-D-23-09655R2 

PLOS ONE

Dear Dr. Muhammad, 

I'm pleased to inform you that your manuscript has been deemed suitable for publication in PLOS ONE. Congratulations! Your manuscript is now being handed over to our production team.

Kind regards, 

on behalf of

Dr. Jin Wu 

Academic Editor

PLOS ONE